# Tunable Fluorescence-Responsive Double Hydrophilic Block Polymers Induced by the Formation of Pseudopolyrotaxanes with Cucurbit[7]Uril

**DOI:** 10.3390/polym11091470

**Published:** 2019-09-09

**Authors:** Xiumin Qiu, Xin Wang, Shengzhen Hou, Jin Zhang, Jing Zhou, Yebang Tan

**Affiliations:** 1School of Chemistry and Chemical Engineering, Key Laboratory of Special Functional Aggregated Materials, Key Laboratory of Colloid and Interface Chemistry, Ministry Education, Shandong University, Jinan 250100, China; 2School of Light Industry and Engineering, Qilu University of Technology, Jinan 250353, China

**Keywords:** supramolecular chemistry, host-guest system, self-assembly, aggregation-induced emission, cell imaging, drug delivery

## Abstract

There is an urgent need for new strategies that allow the simultaneous detection and control of drug delivery. By making use of supramolecular host-guest interactions, a kind of pseudopolyrotaxanes, as a visualizable nanoscale drug carrier has been constructed by self-assembly of cucurbit[7]uril (CB[7]) with methoxy poly(ethylene glycol)-*block*-quaternized poly(4-vinyl pyridine) (mPEG-*b*-QP4VP) using 4-(chloromethyl)benzonitrile. Simple addition of CB[7] into an aqueous solution of mPEG-*b*-QP4VP resulted in noncovalent attachment of CB[7] to 4-cyanobenzyl-containing polymers, transforming the nonemissive mPEG-*b*-QP4VP micelles into highly fluorescent micelles. These pseudopolyrotaxanes micelles exhibited remarkable supramolecular assembly-induced emission enhancement and excellent biocompatibility, showing great potential for bioimaging applications. In addition, the efficient cellular uptake of the developed pseudopolyrotaxanes micelles loaded with the anticancer drug doxorubicin was a promising platform for simultaneous cell imaging and drug delivery, thereby widening their application in cancer theranostics.

## 1. Introduction

In the past decade, polymeric micelle systems have attracted much attention for their potential application in drug carriers. The reasons are that they have many advantages such as nanoscale size, good biocompatibility, good stability, high drug-loading capability, longevity in vivo, and adjustable functionality [1,2,3,4]. In recent years, several approaches have been applied to construct nanoassemblies (particles or micelles), including self-assembly of amphiphilic block copolymers, assembly of macromolecules, and different interactions (electrostatic binding). Välimäki et al. reported a resorcinarene-polymer complexes based on halogen-bonded [5], which is between a halogen bonding acceptor N-benzyl ammonium resorcinarene bromide and a linear iodine functionalized polymers (halogen bonding donors). Nummelin et al. utilized amphiphic janus dendrimers to mediate the formation of fibers and supramolecular hydrogels [6]. In addition, multi-stimuli responsive polymeric micelles, which can respond to the specific microenvironment of tumor cells and tumor tissue, are also desirable [7,8]. Some strategies have been developed to obtain the desirable polymeric micelles, including temperature, pH, redox and electric/magnetic fields [9,10,11,12]. Among these strategies, pH-responsiveness is of the most important, since it is widely accepted that the pH of endosome/lysosome (pH < 5.0) and tumor tissues (pH < 6.8) are slightly acidic [13,14], which is markedly different from normal tissues (pH = 7.4). The hydrophobic-to-hydrophilic transformation of the poly(4-vinyl pyridine) (P4VP) in low pH (pH < 4.7) makes it become more and more popular as a novel environmentally responsive moiety, which endows the system with excellent pH-responsive properties [15,16]. Moreover, the pyridine nitrogen atoms on the P4VP chains can be quaternized by alkyl halides [17,18]. Further targeted functionalization of the polymers can be obtained by simple modification of the P4VP chains. Yao Wu et al. reported a pH-responsive desirable drug delivery carrier. By the co-assembly of the fixed mPEG-*b*-P4VP modified Fe_3_O_4_ NPs (D-Fe_3_O_4_@ mPEG-*b*-P4VP) and different well-designed free mPEG-*b*-P4VP, this endows these hybrid superstructures with multi-functionalization [19]. It is envisaged that similar polymers will have potential applications as intelligent nanomaterials for biomedicine. Through the simple chemical modification of such responsive polymers, functions other than drug delivery can also be attained. It is widely accepted that for conventional nanocarriers, it is usually hard to trace the intracellular drug release after the drug has entered the cancer cells [20]. Therefore, the development of visualizable drug delivery systems (DDS) would be an important achievement. To this end, a number of water-soluble fluorescent polymers have been developed to fabricate DDS [21,22,23,24]. Compared with small-molecular fluorescent probes, water-soluble fluorescent polymers have a wide range of applications in biological detection and diagnosis owing to their better photostability, biocompatibility, and water stability [25,26,27,28,29]. However, conventional water-soluble fluorescent polymers are limited by the aggregation caused quenching (ACQ) effect [30,31,32]. Therefore, effective and convenient strategies have been developed to overcome this defect. Aggregation-induced emission (AIE) is a proven efficient method to fabricate functional aggregates with strong fluorescence [30,33]. Another approach to suppress fluorescence quenching in aqueous solutions is to form host-guest complexes through non-covalent interactions with macrocyclic hosts, such as cyclodextrin (CD) and cucurbit[n]uril [34,35,36,37,38]. However, AIE-based fluorophores often require complex organic synthesis with multiple steps. In contrast, the supramolecular approach offers a convenient and low-cost strategy for smart luminescent materials, because the photophysical properties can be tuned by varying the non-covalent interactions [39]. Recently, supramolecular approaches, such as host-guest interaction, have been widely used to suppress ACQ [40,41,42]. Supramolecular smart materials based on host-guest interactions have become increasingly popular [43,44,45,46]. 

Cucurbit[n]urils (Q[n]s or CB[n]s), a family of a new class of synthetic macrocyclic host molecules, have drawn increasing attention in supramolecular chemistry and materials science [47,48]. Their popularity stems from their good selectivity and strong binding affinity with a range of guest molecules in aqueous solution [49,50,51,52]. Among CB[n]s, CB[7] presents high water solubility (ca. 5 mM) and a favorable cavity size. The outstanding properties of CB[7] make it widely employed in the binding of fluorescent dyes [42]. Recent reports on the interactions of CB[7] with fluorophore guest molecules revealed that CB[7] results in a significant modification of the physicochemical properties of dyes upon complexion [53,54,55,56,57,58]. However, these efforts have focused on the encapsulation of small dyes by CB[7] [53]. At present, no interaction studies have been conducted on dye-labeled double hydrophilic block polymers and CB[7]. Moreover, there are very few reports of using host-guest recognition motifs to fabricate visualizable drug delivery systems [59,60,61]. Therefore, we fabricated a fluorescence pseudopolyrotaxanes nanocarrier by self-assembly of 4-cyanobenzyl-conjugated methoxy poly(ethylene glycol)-*b*-poly(4-vinyl pyridine) (mPEG-*b*-QP4VP) copolymer and CB[7]. The mPEG-*b*-QP4VP copolymer self-assembled into a core-shell structure in water, and the weak fluorescence intensities of the micelles were observed, owing to the aggregating of the 4-cyanobenzyl group in the core of the micelles (micelle-induced fluorescence quenching). Interestingly, the simple addition of CB[7] into an aqueous solution of mPEG-*b*-QP4VP not only effectively reduced the toxicity of the mPEG-*b*-QP4VP, but also prevented the fluorescence quenching of the 4-cyanobenzyl group in the micelles. These properties endowed these micelles with cytocompatibility and were suggestive of potential applications for cell imaging and tracking intracellular drug delivery.

## 2. Materials and Methods

### 2.1. Materials

Methoxy poly (ethylene glycol) (mPEG; *M*_w_ = 1900 g/mol, Alfa Aesar (Shanghai, China), 99%), Tri[2-(dimethylamino)-ethyl]amine (Me_6_TREN), 4-(chloromethyl)benzonitrile, and 4-vinylpyridine (4VP) were purchased from J&K Scientific (Beijing, China). CuBr (97%, Sigma-Aldrich, Shanghai, China) and (4VP, 96%) were purified according to a previously reported procedure [62]. Bromide-tail macro initiator mPEG-Br (Appendix A) was synthesized as described previously [62,63]. The self-assembly process of the core-shell micelle was prepared by mixing the mPEG-*b*-QP4VP solution with CB[7] suspension and pure water. The 20 mg mPEG-*b*-QP4VP was dissolved in deionized water (10 mL), and 2.0 mL (34 mg/mL) CB[7] suspension was added. The mixture was mixed with vortex (2500 rpm, 30s). Then the system was dialyzed against a large amount of pH = 7.4 PBS buffer using a dialysis membrane (MWCO = 3000 Da) for 24 h changing the water every 6 h, and the self-assembled micelles formed. All solvents were dried and redistilled before use.

#### 2.1.1. Preparation of mPEG-*b*-P4VP

Amphiphilic methoxy poly(ethylene glycol)-*block*-poly(4-vinylpyridine) (mPEG-*b*-P4VP) was synthesized by atom-transfer radical polymerization (ATRP). The procedure for the synthesis of mPEG-*b*-P4VP was as follows: the mixture of mPEG-Br (1.025 g), 4VP (5.26 g, 0.05 mol), and Me_6_TREN (0.230 mL, 1 mmol) were dissolved in 3 mL of solvent (a 2:1 mixture of butanone and 2-propanol by volume) and was degassed by three-pump-thaw cycles, and then CuBr (0.072 mg, 0.5 mmol) was added under the nitrogen atmosphere. The mixture was subjected to two more freeze-pump-thaw cycles, and the final reaction mixture was placed in an oil bath at 45 °C for 4 h (Appendix A). Subsequently, the reaction was exposed to air and the suspension was diluted with CH_2_Cl_2_ (10 mL). The purified polymer was obtained via passing through a neutral alumina column chromatography and precipitation in cold ether. mPEG-*b*-P4VP was precipitated with cold diethyl ether three times. mPEG-*b*-P4VP (3.99 g) was obtained with a yield of 63.48%.

#### 2.1.2. Quaternization of P4VP Block in mPEG-*b*-P4VP

mPEG-*b*-P4VP (0.50 g) and C_2_H_5_OH (15 mL) were introduced into a 50 mL flask A with gentle stirring. Meanwhile, 4-(chloromethyl)-benzonitrile (1:1 ratio to the number of 4VP units) was added into flask B in C_2_H_5_OH (10 mL). Subsequently, the solution in flask B was added slowly into flask A. The final mixture was placed in an oil bath at 45 °C for 24 h (Appendix A). The unreacted 4-(chloromethyl) benzonitrile was removed through dialysis with a 3000 molecular weight cut off (MWCO) semi-permeable membrane in C_2_H_5_OH for 3 days. The purified product was obtained via concentration and precipitation from C_2_H_5_OH into ether (0.583 g, yield 81.79%).

#### 2.1.3. Doxorubicin-Loading and Controlled Release from the Micelles

DOX·HCl was dissolved in *N*,*N*-dimethylformamide, and then the solution was neutralized with triethylamine to form hydrophobic doxorubicin. Subsequently, DOX (2 mg) and pseudopolyrotaxanes (20 mg) were dissolved in DMF (4 mL) and H_2_O (5 mL), respectively. The above DOX solution (0.5 mg/mL) was added slowly to the pseudopolyrotaxanes solution with stirring. After 4 h stirring, a purple-blue homogeneous solution was obtained. The resulting solution was dialyzed against PBS for 24 h in a dialysis bag (MWCO 3000). The external buffer solution was changed every 4 h to remove free DOX molecules. Then, the solution was centrifuged and lyophilized. Drug-loading efficacy (DLE) was analyzed by using UV spectroscopy in accordance with the literature [64,65]. The blank micelles of the pseudopolyrotaxanes were prepared in the same way as the DOX-loaded pseudopolyrotaxanes micelles, but without the addition of the drug.

The drug loading efficiency (DLE %) of DOX was calculated from:DLE (%) = (mass of DOX in micelle/mass of total DOX added) × 100%
DEE (%) = (mass of DOX in micelle/mass of DOX in feeding) × 100%.

For the DOX release experiment, the freeze-dried micelles loaded with DOX were dispersed in PBS (pH 7.4) buffer at a concentration of 1.0 mg/mL. The in vitro drug release was studied at 37 °C in PBS with pH 7.4, pH 5.0, and pH 4.5. Briefly, 1.0 mg/mL of DOX-loaded micelles was injected into dialysis membrane tubes (3000 MWCO) with 30 mL buffer solution as the medium. The above system was then incubated at 37 °C in the dark with shaking. At prescribed time intervals, 3 mL sample was take out and replaced by 3 mL of fresh buffer solution. The released DOX was measured by using a fluorescence spectrometer. The release experiments were performed in triplicate, and the results were expressed as the mean ± SD.

#### 2.1.4. In Vitro Cell Viability Assay 

The cytotoxicity of DOX·HCl, mPEG-*b*-QP4VP, pseudopolyrotaxanes, and DOX/pseudopolyrotaxanes assembly were assessed in vitro by an MTT assay. NIH3T3 mouse fibroblasts cells and HeLa cells were separately inoculated into 96-well plates at a density of 5 × 10^4^ cells per well in 100 μL of medium and incubated for 24 h. The cells were then treated with DOX·HCl, mPEG-*b*-QP4VP, pseudopolyrotaxanes, and DOX/pseudopolyrotaxanes at given concentrations. After incubation for 24 or 48 h, the medium was discarded and the cells were rinsed twice with PBS (pH 7.4). Subsequently, the MTT solution (10 μL of 5 mg/mL in PBS (pH 7.4)) was added to each well and the plates were incubated for another 4 h. The optical density (OD) of the solution in each well was measured in comparison with a blank well (containing only DMSO) by using an EnSpire (PerkinElmer, Boston, MA, US) plate reader at a wavelength of 490 nm. The untreated cells were used as the positive control. All experiments were conducted in triplicate (mean ± SD, *n* = 3).

The relative cell viability was calculated from the following equation:Cell viability (%) = (OD_treated_ − OD_MTT_/OD_control_ − OD_MTT_) × 100%
where OD_treated_ was the value obtained in the presence of the micelles and OD_control_ was the value obtained in the absence of the micelles.

#### 2.1.5. Cell Uptake and Intracellular Drug Released 

The cell uptake and drug distribution of the DOX-loaded micelles were determined by confocal laser scanning microscopy (CLSM). For CLSM studies, HeLa cells were incubated on 35 mm diameter glass dishes at a cell density of 1 × 10^4^ mL^−1^. Subsequently, cells were incubated with DOX·HCl and DOX-loaded micelles at a final concentration of 10 μg mL^−1^ for 4 h at 37 °C. The culture medium was removed and the cells were rinsed three times with PBS. Subsequently, the images were captured by using a Leica TCP SP8 (Leica, Wetzlar, German).

### 2.2. Characterization

^1^H NMR spectra were collected by using a Bruker Advance III 400 MHz spectrometer (Bruker, Geman) at 298 K; TMS was used as the internal reference. The infrared spectra were recorded by using a Bruker Vertex70 spectrometer (Bruker, Geman) working in attenuated total reflectance mode, the scan range was 4000–400 cm^−1^. The polymer number-average molecular weight and distribution (also polydispersity index, PDI) was identified by gel permeation chromatography (GPC) with DMF as an eluent (Waters, Massachusetts, Milford, US). UV-vis spectra were measured by using a Persee TU1901 (Persee, Beijing, China) ultraviolet spectrophotometer. Fluorescence spectra were recorded by using a Hitachi F-7000 (Hitachi, Tokyo, Japan) fluorescence spectrometer. Isothermal titration calorimetry (ITC) experiments were performed at 298 K by using a Microcal VP-ITC (Malvern Instruments GmbH, Malvern, England) apparatus at 298 K. Dynamic light scattering (DLS) experiments were performed on a Malvern Zetasizer Nano ZS (Malvern Instruments GmbH, Malvern, England). Samples were filtered through a MILLEX-GV 0.45 µm filter before loading. Transmission electron microscopy (TEM) experiments were performed on a Hitachi-7700-MS electron microscope (Hitachi, Tokyo, Japan). The quantitative evaluation of the cell uptake and release were performed on a BD FACSAria III flow cytometer (BD, San Jose, CA, US).

## 3. Results

### 3.1. Synthesis and Characterization of mPEG-b-P4VP and mPEG-b-QP4VP

The methoxy poly(ethylene glycol)-*block*-poly(4-vinyl pyridine) (mPEG-*b*-P4VP) block polymer was synthesized through ATRP using mPEG-Br as the macroinitiator (Appendix A). The synthesized mPEG-*b*-P4VP copolymers were confirmed by ^1^H NMR and FT-IR spectra (Appendix A). In the ^1^H NMR spectrum (Appendix A), the chemical shift of the P4VP characteristic protons was at 1.10–2.00 ppm (d, 3H), the chemical shift of mPEG backbone was at 3.30–3.70 ppm (b, 4H), and the aromatic protons was at 6.30–6.90 ppm (e, 2H) and 7.80–8.5 ppm (f, 2H), confirming the chemical structures of the synthesized copolymers. Subsequently, mPEG-*b*-P4VP was partially quaternized by using 4-(chloromethyl)benzonitrile in C_2_H_5_OH. The obtained copolymer was characterized by ^1^H NMR (Appendix A). By the quaternization of the P4VP block, a new characteristic signal was observed. Signals that appeared at 5.40–5.90 ppm (d, 2H) were attributed to the protons of CH_2_ in the 4-cyanobenzyl group. Moreover, the characteristic protons signal of P4VP at 6.19–6.62 ppm and 8.14–8.51 ppm shifted to 7.50–7.90 ppm and 7.80–8.70 ppm, respectively. In addition, the degree of quaternization of the 4VP units was also estimated by the ^1^H NMR spectrum (Appendix A). According to the ^1^H NMR spectrum, we can see that only approximately 40% of 4VP units in the polymer chain were quaternized. This could be attributed to steric hindrance and crowded environment in the polymer [18,66]. The partial quaternization of P4VP block with 4-cyanobenzyl group not only renders the polymer well fluorescence properties, but also maintains the pH-responsiveness of the P4VP block. Therefore, it is possible to study the fluorescence and pH dual-responsiveness of the diblock copolymer. The FT-IR specta of mPEG, mPEG-Br, mPEG-*b*-P4VP, and quaternized copolymer mPEG-*b*-QP4VP are shown in Appendix A. In the FT-IR spectra, other than the three absorption bands at 1732 cm^−1^ (C=O), 1600 cm^−1^ (C=N), and 1559 cm^−1^, the characteristic absorption of the quaternary N atom was observed at 1638 cm^−1^. The ^1^H NMR spectrum combined with the FT-IR spectrum of mPEG-*b*-QP4VP (Appendix A), which illustrated the successful synthesis of the desired block copolymers. According to the ^1^H NMR, the degree of polymerization (DP) of the hydrophobic P4VP block was calculated to be 103 based on the ^1^H NMR spectrum. The estimated number-average molecular weight (*M*_n_) of the block copolymers based on ^1^H NMR signals was 12864 g/mol. The molecular weight (*M*_n_ = 16226) of mPEG-*b*-P4VP was measured by GPC with the PDI of 1.15 (Appendix A).

The concentration-dependent fluorescence emission spectra of mPEG-*b*-QP4VP in water were measured at an excitation wavelength at 365 nm (Appendix A). The results given in the fluorescence spectra revealed that the intensity of the emission at 432 nm greatly increased as the concentration of mPEG-*b*-QP4VP was increased from 0.01 mg/mL to 35 mg/mL, whereas at the high mass concentration (≥35 mg/mL), the emission intensity greatly decreased as the concentration of the copolymer increased. The formation of clusters provided a possible explanation for the observed change in the fluorescence spectra [67]. The fluorescence spectrum at approximately 432 nm showed the presence of a low concentration of the 4-cyanobenzyl moiety. At the low concentration (35 mg/mL), the intensity of the peak at 432 nm greatly increased as the concentration of the copolymer increased, implying that the clustering did not occur in the solution. In contrast, the clustering occurred and the fluorescence was quenched at high mass concentrations (35 mg/mL). In addition, as the polymer concentration decreased, the emission intensity tended to decline, indicating that a high concentration of mPEG-*b*-QP4VP (≥0.05 mg/mL) in an aqueous solution could produce a detectable signal suitable for optical imaging. The pH responsive of the mPEG-*b*-P4VP micelles is well discussed in the Appendix A (Appendix A).

### 3.2. Binding Behavior of mPEG-b-QP4VP with CB[7]

^1^H NMR spectroscopy can be used to study polymer structural properties and molecular interactions. The ^1^H NMR titration spectra of the mPEG-*b*-QP4VP in D_2_O recorded with the increasing amounts of host CB[7] are shown in Figure 1. Upon the gradual addition of the CB[7] to an aqueous solution of mPEG-*b*-QP4VP, the signals of the aromatic protons (H_d_ and H_e_) on the 4-cyanobenzyl group were shifted dramatically upfield. Meanwhile, when 1 equiv of CB[7] was added to the mPEG-*b*-QP4VP solution, the signals assigned to the 4-cyanobenzyl group of the copolymer at 7.82 and 7.42 ppm disappeared (see Figure 1D). It was suggested that these signals were shifted upfield based on the spectrum of the 4-cyanobenzyl group CB[7] host-guest complexes. The signal of CB[7] also changed significantly after the formation of the host-guest complexation. The methylene hydrogen on CB[7] was also split and broadened (peak at 4.17 ppm; Figure 1C) [68]. These observations implied that the CB[7] may wrap around the 4-cyanobenzyl group of the copolymer and the formation of pseudopolyrotaxanes. In summary, the ^1^H NMR titration experiments revealed that when the host is in stoichiometric (CB[7]:4-cyanobenzyl = 1:1), a full encapsulation of the 4-cyanobenzyl group in the cavity of the CB[7] host was observed.

We further explored the complexation behavior of CB[7] with mPEG-*b*-QP4VP by ITC experiments. As shown in Figure 2, the obtained data were analyzed according to the binding models that were given in the ITC tutorial guide, and were found to fit a single binding model. The association constant (Ka) was determined as (1.65 ± 0.097) × 10^5^ M^−1^ (Figure 2). The ITC results confirmed the 1:1 binding between CB[7] and 4-cyanobenzyl group in mPEG-*b*-QP4VP.

In an effort to gain more detailed information on the host-guest interactions between CB[7] and mPEG-*b*-QP4VP, the inclusion event was thoroughly investigated through analysis of the UV-vis spectra and fluorescence emission spectra. The UV-vis absorption spectral changes in mPEG-*b*-QP4VP as a function of added CB[7] are shown in Appendix A. The parent polymer mPEG-*b*-QP4VP exhibited a weaker absorption centered at 229 nm and 256 nm, which are characteristic absorption peaks of pyridinium and the 4-cyanobenzyl group (π–π*). Upon the addition of CB[7] to an mPEG-*b*-QP4VP solution, the intensity of the band at 229 nm and 256 nm increased. However, the increase was not evident, which may relate to the formation of a hyper-conjugated system between the C=O of CB[7] and the N^+^ of the pyridinium salt [69]. In addition, both the 4-cyanobenzyl group and CB[7] could form the pseudopolyrotaxanes through host-guest interactions, and the 4-cyanobenzyl group was shielded by the inclusion of CB[7], which agreed with the ^1^H NMR spectrum [70]. When the molar ratio of CB[7] : mPEG-b-QP4VP increased from 1:0 to 1:1, the fluorescence emission intensities increased gradually (Figure 3B). Meanwhile, an enhancement of approximately two-fold was calculated to occur after the addition of CB[7] by a stoichiometric ratio (Host:guest = 1:1). Moreover, the increase in emission intensity became slower when one equivalent of CB[7] was introduced, indicating the formation of a 1:1 inclusion complex (Figure 3B). In addition, the absolute fluorescence quantum yield (Φ_f(abs)_) of mPEG-*b*-QP4VP increased from 5.07% to 6.90%. The decay lifetime of the mPEG-*b*-QP4VP and pseudopolyrotaxanes were 2.068 ns and 2.310 ns, respectively (Appendix A). The results further indicated intermolecular interactions between CB[7] and mPEG-*b*-QP4VP. It was estimated that the fluorescence of the mPEG-*b*-QP4VP assemblies was severely quenched owing to the close aggregation of the 4-cyanobenzyl group in the micelles. However, the formation of host-guest complex between CB[7] and 4-cyanobenzyl group suppressed the π–π interactions and fluorescence quenching of the 4-cyanobenzyl aggregates, leading to the fabrication of highly fluorescent assemblies.

As amphiphilic copolymers, the mPEG-*b*-QP4VP self-assembled into micelles with P4VP as the core and biocompatible PEG as the shell at 25 °C in an aqueous solution. 

The properties of the pseudopolyrotaxanes were similar to those of the mPEG-*b*-QP4VP, as shown in Scheme 1a. DLS and TEM studies were performed to study the size and morphology of these micelles. As illustrated in Figure 3F, the hydrodynamic radius (*R*_h_) of the mPEG-*b*-QP4VP increased with an increase in the CB[7] content. It should be noted that the size of the micelles that self-assembled did not significantly increase, and the largest amplitude was from 92 nm to 104 nm. The results were similar to our previous reports [71]. Furthermore, the morphology of mPEG-*b*-QP4VP and pseudopolyrotaxanes was observed by TEM. As is clearly shown in the TEM images (Figure 3E), the micelles generally had a regular spherical morphology with an average radius of approximately 50 nm. The size of the self-assembled units increased from 50 nm to 69 nm as the CB[7] content of the pseudopolyrotaxane units increased. The size observed by DLS was larger than that determined by TEM. This difference was attributable to the fact that DLS measurements were in an aqueous solution, whereas the TEM images were observed after the solvent evaporation, which may lead to the shrinkage, collapse, or destruction of micelles [72]. Therefore, to some extent, the results of TEM are consistent with those of the DLS measurements. 

### 3.3. Drug Release Profiles

As pseudopolyrotaxanes micelles are multifunctional and cytocompatible, they may be used as a smart component in a drug carrier. DOX was chosen as a model drug to investigate the controlled release properties of the micelles. After drug loading, DOX-loaded micelles (122.5 nm) were larger than blank ones (109.2 nm), indicating successful loading of DOX into the micelles (shown in Appendix A). The TEM images in Appendix A show that DOX-loaded micelles unaltered regularly spherical shape, suggesting that the drug loading did not affect the dimension and morphology of the micelles [73]. From the standard curve, the drug loading efficiency was calculated to be 7.2%, suggesting that these micelles were a suitable platform for the loading of a hydrophobic anticancer drug. The release behavior of DOX-loaded micelles was monitored in aqueous solutions of different pHs. As shown in Figure 4B, as the pH of the solution decreased, the release of DOX from the pseudopolyrotaxanes micelles increased: 21% release efficiency at pH 7.4, 51% at pH 5.0, and 70% at pH 4.5 after 72 h, suggesting that the drug release behavior of pseudopolyrotaxanes micelles was induced by the acidic environment. Such strong pH sensitivity was attributed to the gradual protonation of the pyridine groups at acidic pH values, which might result in the destruction of DOX-loaded micelles. In addition, the low pH may also lead to the protonation of DOX, which promoted the escape of water-soluble DOX·HCl from the hydrophobic core of the pseudopolyrotaxanes micelles. This shows potential efficacy for accelerating the drug release and should improve the antitumor effect [74,75]. 

### 3.4. In Vitro Cell Viability Assay 

The cytotoxicity tests of polymeric micelles to NIH3T3 fibroblasts and HeLa cells were evaluated by an MTT assay. As shown in Figure 4A, the viability of NIH3T3 and HeLa cells treated with pseudopolyrotaxanes micelles for 48 h was over 85% in all tests, even at applied concentrations up to 500 μg/mL. The results suggested that the pseudopolyrotaxanes micelles were nontoxic and were suitable for drug delivery. The viability of NIH3T3 cells treated with mPEG-*b*-P4VP and pseudopolyrotaxanes is shown in Appendix A (Appendix A). The efficiency of inhibition of tumor cells by free DOX·HCl and DOX-loaded micelles was also assessed by MTT assay (Figure 5). As shown in Figure 5, free DOX·HCl displayed the most effective inhibition of cell proliferation because the water solubility of DOX·HCl allowed it to diffuse quickly into cells and induce cell apoptosis [1]. The HeLa cell viability decreased to 25% after 24 h and 15% after 48 h. Furthermore, at the lower drug concentration, the DOX-loaded micelles exhibited lower cell proliferation inhibition efficiency of HeLa cells as compared with the free DOX. The slower drug release from the DOX-loaded micelles may be attributable to the enhancement of micellar stability [1]. Meanwhile, the viability of the cells is almost the same at high concentrations, which indicates that the loading of DOX into micelles did not affect inhibition efficiency of the DOX.

### 3.5. Cell Uptake 

Figure 6 shows the cellular uptake and drug distribution of DOX-loaded pseudopolyrotaxanes micelles. The confocal microscopy images of HeLa cells were utilized to detect the endocytosis and intracellular distribution of these micelles. After incubation with DOX·HCl, mPEG-*b*-P4VP, pseudopolyrotaxanes micelles, and DOX/pseudopolyrotaxanes micelles for 4 h, the red fluorescence of DOX was observed in HeLa cells, and was mostly located in the cell nucleus (Figure 6A). Similarly, the weak blue fluorescence arising from mPEG-*b*-QP4VP was also observed in the cell nucleus (Figure 6B). However, the blue fluorescence was too weak to be observed because of the fluorescence of the 4-cyanobenzyl group was severely quenched. This phenomenon was attributable to the energy transfer in the 4-cyanobenzyl group [76]. Interestingly, the blue fluorescence of the pseudopolyrotaxanes micelles was greatly enhanced compared with that of mPEG-*b*-QP4VP micelles. The images showed the formations of sizes of approximately dozens of micrometers, which indicated that the pseudopolyrotaxanes micelles were nontoxic and could be suitable for drug delivery (Figure 6C). The formation of CB[7] pseudopolyrotaxanes micelles not only effectively reduced the toxicity of mPEG-*b*-QP4VP, but also prevented fluorescence quenching in the micelles. Then, the intense blue fluorescence signal was detected from the cell nucleus. As can be seen in Figure 6D, the red fluorescence of DOX together with the blue fluorescence of mPEG-*b*-QP4VP was observed simultaneously in the nucleus of HeLa cells; it is implied that pseudopolyrotaxanes assemblies would be used as a visible carrier in cells and might have potential application in tracing the drug distribution. A similar tendency was further illustrated by flow cytometry analyses (as shown in Figure 7). As can been seen in Figure 7A, compared with mPEG-*b*-QP4VP micelles, cells co-cultured with pseudopolyrotaxanes micelles showed stronger fluorescence intensity, which was in accordance with the CLSM results. In addition, the fluorescence intensity of DOX-loaded pseudopolyrotaxanes micelles was similar to DOX·HCl at a high DOX concentration, which indicated that the internalization of the DOX-loaded micelles was comparable to DOX·HCl under the same conditions (Figure 7B). All the results manifested that the pseudopolyrotaxanes micelles were highly effective in drug delivery and cancer theranostics.

## 4. Conclusions

In summary, fluorescence pseudopolyrotaxanes nanocarriers were developed for simultaneous cellular imaging and drug delivery. The nanocarriers were fabricated via the self-assembly of dye-labeled double hydrophilic block polymers and CB[7]. The pseudopolyrotaxanes self-assembled into micelles in an aqueous solution and the drug, DOX, was loaded into the micelles. Through the simple addition of CB[7] into an aqueous solution of mPEG-*b*-QP4VP, the bulky CB[7] was noncovalently attached to the 4-cyanobenzyl group in mPEG-*b*-QP4VP, which reduced the toxicity of mPEG-*b*-QP4VP and enhanced the fluorescent intensity of mPEG-*b*-QP4VP micelles. The noncovalent interactions between the 4-cyanobenzyl group and CB[7] resulted in the property of polymeric micelle supramolecular assembly-induced emission enhancement. Therefore, the supramolecular approach provides a convenient and efficient avenue for fabricating theranostic nanocarriers, which are promising for imaging-guided anticancer drug delivery.

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
