# Peer review of "Tunable Fluorescence-Responsive Double Hydrophilic Block Polymers Induced by the Formation of Pseudopolyrotaxanes with Cucurbit[7]Uril"

_polymers, 2019, doi:10.3390/polym11091470_

Round 1

Reviewer 1 Report

Qiu et al. present a method to form micelle-like structures using a host-guest chemistry and hydrophilic block polymers. The idea to form carriers that are suitable for imaging while delivering a cargo is good. However, there are several issues, and in my opinion, the manuscript may become publishable after major revision. The following concerns should be thoroughly addressed, please see below:

- In the introduction, I suggest the authors should mention other strategies for forming nanoassemblies (particles or micelles) using synthetic polymers and different (but non-covalent) interactions. See for example resorcinarene-polymer complexes through halogen-bonding (e.g. S. Välimäki et al. Macromol. Rapid. Commun. 2019, 40, 1900158) and Janus-dendrimer-based dendrimersomes (e.g. S. Nummelin et al. Chem Eur. J. 2015, 21, 14433).

- Figure 3: The authors should also present the correlation curves of the DLS experiments. By zooming in, it can be seen that PDI values are 0.2 or less, which is within acceptable range. However, this text in figure 3E is way too small, it should be presented in a clearer way. 

- DOX-loading: How specific is the DOX-loading? The loading efficiency depends also on pH, just as the release. Have the authors tested the loading at different pH values?

- Please include the sample sizes to each figure caption (N = ?) and also report how the error bars have been defined (standard deviation?)

- The sentence: "the DOX-loaded micelles exhibited lower cell proliferation inhibition efficiency of Hela cells as compared with the free DOX." is a bit misleading. The viability of cells are almost the same at high concentrations.
Moreover, there is not much difference between 24 and 48 h cases, which indicates that the loading of DOX into micelles is not affecting that much (no sustained release, for example).

- Cell uptake: the authors should use FACS or similar method to get quantified data of the cell uptake.

- Confocal imaging: have the authors tried different time scales? Does more than 4 h incubation cause cell death?

Reviewer 2 Report

In the manuscript “Tunable fluorescence-responsive double hydrophilic block polymers induced by the formation of pseudopolyrotaxanes with cucurbit[7]uril” the authors describe the synthesis of quaternized poly(ethylene glycol)-b-poly(4-vinylpyridine) (mPEG-b-QP4VP) and their supramolecular assembly to micelles with cucurbit[7]uril. The quaternised polymer itself exhibits very weak luminescent properties due to the aggregation of the chlorobenzyl moiety within the core of the core-shell particles leading to the aggregation caused quenching (ACQ). The authors took advantage of supramolecular host-guest interaction in order to suppress the ACQ effect. The straightforward addition of the macrocyclic cucurbit[7]uril host molecule allows to enclose the 4-cyanobenzyl guest molecule, which resulted in the elimination of the ACQ effect. The now fluorescent system can be used for imaging applications as well as a tracer for drug delivery.

In this study, the group has shown, that the emissive behavior of the micellar system can be controlled by tuning the amount of added host molecules with respect to the amount of guest molecules. Apart from the inherent good water solubility of CB[7] and the suitable cavity for the cyanobenzyl group, the assembled polymeric system exhibited an reduced cytotoxicity after supramolecular complexion. Doxorubicin was also loaded onto the micelles which was released into cells due to the pH-responsiveness of the 4VP units. Currently, there is a plethora of studies regarding encapsulation of small dyes using supramolecular host-guest interactions with cucurbiturils as host, but there are less studies regarding cucurbituril and dye functionalized polymers systems. I recommend publication of this manuscript after addressing some minor comments and questions.

Minor coomments:

pH-responsiveness: I agree with the authors that P4VP is pH-responsive and that the phenomenon is discussed in detail in the SI. However, after quaternization (40% is typical), the Q4VP becomes a strong polyelectrolyte and is usually soluble over the entire range of pH range. Do the authors have an alternative explanation for the assembly at higher pH (other than collapsing P4VP)?

In line 95: Although the quaternization of the block copolymer and the DOX-loading into micelles was described in the Material and Methods section, there are no experimental details about the self-assembly process of the core-shell micelle themselves. Please provide a description about how micelles were assembled.

In line 180: for the final block copolymer mPEG-b-P4VP the number of repeating units is not given anywhere. Please use NMR to calculate the block ratios. The molecular weight of the GPC is not sufficient as the calibration curve is not suitable for the diblock. How was the amount of 4-(chloromethyl)benzonitrile or curcurbit[7]uril calculated without determining the block length of P4VP?

In lines 259-262: It is mentioned that the emission intensities increase gradually by adding CB[7] until the formation of a full inclusion complex (1:1 ratio of CB[7] and 4-cyanobenzyl group). Comparing cuvettes d and e in Figure 3C, it seems that the intensity is still increasing although cuvette e already reaches a ratio of 1:2 guest/host. How does the intensity further increase, when all benzyl groups are already inside a cavity?

Round 2

Reviewer 1 Report

The authors have addressed my main concerns, and therefore I think the manuscript could now become publishable.